# Meeting the Challenges of the UN Sustainable Development Goals through Holistic Systems Thinking and Applied Geospatial Ethics

Christy M. Caudill *, Peter L. Pulsifer, Romola V. Thumbadoo and D. R. Fraser Taylor

Geomatics and Cartographic Research Centre (GCRC), Carleton University, 1125 Colonel By Dr HC 3110, Ottawa, ON K1S 5B6, Canada; ppulsifer@gcrc.carleton.ca (P.L.P.); rvthumbadoo@gcrc.carleton.ca (R.V.T.); frasertaylor@cunet.carleton.ca (D.R.F.T.)
* Correspondence: ccaudill@gcrc.carleton.ca

**Abstract:** The halfway point for the implementation of the United Nations Sustainable Development Goals (SDGs) was marked in 2023, as set forth in the 2030 Agenda. Geospatial technologies have proven indispensable in assessing and tracking fundamental components of each of the 17 SDGs, including climatological and ecological trends, and changes and humanitarian crises and socio-economic impacts. However, gaps remain in the capacity for geospatial and related digital technologies, like AI, to provide a deeper, more comprehensive understanding of the complex and multi-factorial challenges delineated in the SDGs. Lack of progress toward these goals, and the immense implementation challenges that remain, call for inclusive and holistic approaches, coupled with transformative uses of digital technologies. This paper reviews transdisciplinary, holistic, and participatory approaches to address gaps in ethics and diversity in geospatial and related technologies and to meet the pressing need for bottom-up, community-driven initiatives. Small-scale, community-based initiatives are known to have a systemic and aggregate effect toward macro-economic and global environmental goals. Cybernetic systems thinking approaches are the conceptual framework investigated in this study, as these approaches suggest that a decentralized, polycentric system—for example, each community acting as one node in a larger, global system—has the resilience and capacity to create and sustain positive change, even if it is counter to top-down decisions and mechanisms. Thus, this paper will discuss how holistic systems thinking—societal, political, environmental, and economic choices considered in an interrelated context—may be central to building true resilience to climate change and creating sustainable development pathways. Traditional and Indigenous knowledge (IK) systems around the world hold holistic awareness of human-ecological interactions—practicable, reciprocal relationships developed over time as a cultural approach. This cultural holistic approach is also known as Systemic Literacy, which considers how systems function beyond "mechanical" aspects and include political, philosophical, psychological, emotional, relational, anthropological, and ecological dimensions. When Indigenous-led, these dimensions can be unified into participatory, community-centered conservation practices that support long-term human and environmental well-being. There is a growing recognition of the criticality of Indigenous leadership in sustainability practices, as well as that partnerships with Indigenous peoples and weaving knowledge systems, as a missing link to approaching global ecological crises. This review investigates the inequality in technological systems—the "digital divide" that further inhibits participation by communities and groups that retain knowledge of "place" and may offer the most transformative solutions. Following the review and synthesis, this study presents cybernetics as a bridge of understanding to Indigenous systems thinking. As non-Indigenous scholars, we hope that this study serves to foster informed, productive, and respectful dialogues so that the strength of diverse knowledges might offer whole-systems approaches to decision making that tackle wicked problems. Lastly, we discuss use cases of community-based processes and co-developed geospatial technologies, along with ethical considerations, as avenues toward enhancing equity and making advances in democratizing and decolonizing technology.

**Keywords:** capacity exchange; knowledge exchange; equity and diversity narratives; cartography; cognitive justice; social justice; systems thinking; remote sensing; UN sustainable development goals

## 1. Introduction

In 2015, United Nations Member States adopted the 17 Sustainable Development Goals (SDGs) as a strategy to implement recommendations for global prosperity, environmental and social justice, and the equitable redistribution of opportunity for advancement based on 231 unique social-ecological indicators spread across 169 targets. The UN Global Sustainable Development Report 2019—"The Future is Now: Science for Achieving Sustainable Development"—concluded that, despite initial efforts, the world is not yet on track for achieving most of the SDG targets. The halfway mark to meet implementation of the SDGs as set forth in the 2030 Agenda was met with a dire report from the Global Assessment Report on Disaster Risk Reduction, which reads as follows: "despite commitments to build resilience, tackle climate change and create sustainable development pathways, current societal, political and economic choices are doing the reverse" [1]. Many have argued that adjustments to existing policies and attempting market-driven changes are inadequate or ill-positioned to deliver the necessary changes in how people, globally dominant cultures and consequently, global systems, sustainably relate to and draw benefits from the environment [2–5]. In this paper, we will discuss how holistic systems thinking—societal, political, environmental, and economic choices are never separate, but considered a system—might offer approaches to decision making that tackle these wicked problems (e.g., [6]). We present a theoretical framework and describe practical approaches that have been successfully implemented with local communities. A growing body of evidence supports that bottom-up, community-based projects are those that immediately affect the lives of people, and this bottom-up approach has an aggregate effect toward macro-economic and global environmental goals (e.g., [3,7]). The focus on enabling bottom-up impact suggests a shift in focus away from data-centric and policy-driven approaches toward justice and equity drivers, including factors that immediately impact lives, livelihood, and land. Holistic knowledge that can accommodate these different dimensions includes a whole-systems approach but also long-term understandings of these aspects and their relationality [2].

Long before scientific inquiry was formalized, Indigenous peoples around the world developed, transmitted, and applied rich interrelation knowledge from direct experience with human-ecological interactions over millennia [8–10]. Indigenous knowledge (IK) encompasses diverse conceptualizations around the world, but it is generally considered to be a place-based body of knowledge. Indigenous ways of knowing are considered sciences in their own right, [11,12], and IK is distinct from local and citizen science in that it is based in observation as well as interaction with entire ecosystems underpinned by cultural and spiritual values that guide human-ecological relationships [6,10]. Non-Indigenous science, often referred to as "Western" or simply "science" now recognizes Indigenous leadership in sustainability practices and that partnerships with Indigenous peoples are a missing link to approach global ecological crises [13]. A growing body of research calls for a greater recognition of IK, citizen science, and community engagement as valuable tools for policy in ecosystem management [14] and environmental and cognitive equality [15].

This paper is written from a North American settler ("Western") perspective, and we situate ourselves as academics who interact with IK holders in the context of research. In an effort to avoid the pitfalls of postmodern extreme relativism, we acknowledge that the use of the term "Western" risks a blanket categorization of researchers; many researchers who follow Western scientific traditions do not align with anthropocentric, strictly reductionist, or objectivist imperatives (see [16]), and these norms are contrary to some practitioners of inductive and emergent research analysis techniques, such as Grounded Theory [17]. We therefore use the term "Western" in this paper to acknowledge the colonial heritage of our knowledge system, without necessarily suggesting that all science methodologies or

practitioners are similarly aligned. This paper will thus contrast aspects of Western and Indigenous conceptualizations (further described in Section 2.2) in the context of subjects and objects in socio-technological systems, e.g., [18].

A key intent of this research is to acknowledge the equal salience of place-based IK in whole-systems approaches to meet the complex challenges presented in the SDGs. As Western researchers, we couch our own understanding of holistic systems thinking in a Western framework of cybernetics (e.g., [19–22]) and build on the seminal formalization of *Geocybernetics* by Reyes et al. [23,24]. Geocybernetics is described as a transdisciplinary approach that transcends purely empirical research to involve society-wide actors. This approach integrates knowledge and geospatial information, "to support a wide variety of activities that include, among others, better articulation of public policies, improvement of ecosystemic or environmental services that provide support, regulation, cultural provision, and understanding and natural conditions and processes, as well as activities that address problems such as poverty, land use and ownership, sustainability, deforestation, food, health, sustainable food, public safety, risks and vulnerability, business investments..." [23]. We will discuss cybernetics as a potential bridge of understanding across Western and IK systems to, primarily, begin these conversations as acts of settler allyship in geospatial science. We furthermore suggest that building bridges between knowledge systems and conceptual domains provide an inclusive structure to move toward implementation of the SDGs. Although we situate this work within these spheres, we recognize that there have been many attempts to understand and link Western and IK frameworks, including the following: Fikret Berkes' Sacred Ecology [25]; Ray Barnhardt and Anagayuqaq Oscar Kawagley's Culture, Chaos and Complexity [26]; and Elder Albert Marshall's Two-Eyed Seeing [27]. In using systems thinking frameworks, it is our hope that this paper might foster conversations specific to meeting cognitive inequalities between knowledge systems, address the ethical and equitable use of geospatial data, tools, and processes, and enable holistic reconceptualizations of wicked problems. The relationships between the conceptual spheres discussed in this paper are graphically described in Figure 1.

As non-Indigenous scholars, we hope that this paper serves to foster informed, productive, and respectful dialogues so that the strength of diverse knowledges might offer whole-systems approaches to decision making that tackle wicked problems. We hope that by giving due credit to the Indigenous-led decolonization of AI and other digital technologies described in this paper, and amplifying the ideas of other such historically marginalized authors in the field of systems thinking and beyond, we act as allies for a more just and inclusive technological and discursive space. We acknowledge that this advocacy may be both helpful and harmful, and thus, we strive to be self-reflective to ameliorate reinforcing Western-based institutional knowledge at the expense of other ways of knowing. As a matter of ethics and responsibility, and in a conscious attempt to minimize potential risks, we work to continually recognize the historical and ongoing negative consequences to modernization, from which digital technologies are not easily separated. These risks are explored in more depth in Section 3.3. In this paper, we suggest that cybernetics as a Western systems thinking approach can act as one row, or path, honoring the Two Row Wampum treaty relationship. The Two Row Wampum is one of the oldest treaty relationships between the Onkwehonweh (original people) of Turtle Island (North America) and European immigrants and is symbolized by two paths or two vessels traveling down the same river together [28]. In the framing of the Two Row Wampum, cybernetics and Indigenous relational systems might act as two approaches that "travel the river together, side by side, but in our own boat," with neither trying to "steer the other's vessel" (e.g., [29]).

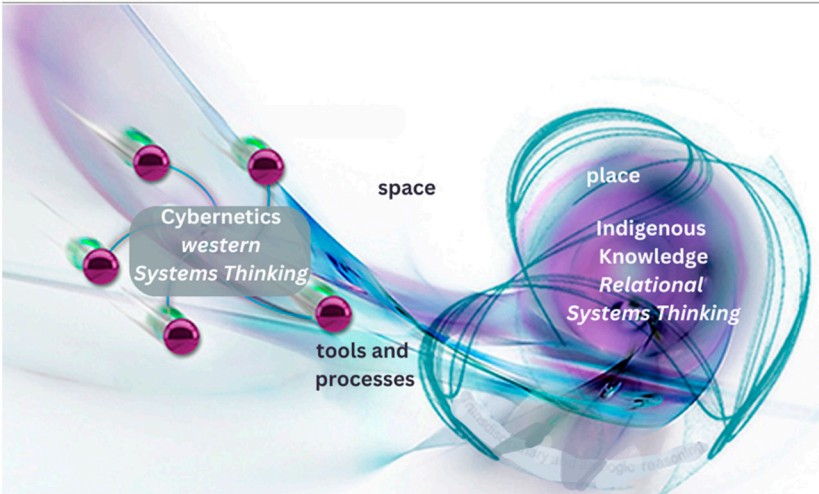

**Figure 1.** Holistic systems thinking approaches are presented in two epistemological spheres as holistic reconceptualizations used to meet the inherent complexities presented in shared global challenges, as those described in the UN Sustainable Development Goals. "Cybernetics" is presented in this paper as a Western systems thinking approach, and the approach of the authors. Separate spheres extend from the Western approach, illustrating reductionism that delineates dimensions (e.g., fields of science; ecology; political realms), with interrelatedness illustrated in a "hub and spoke" configuration. The figure shows an Indigenous knowledge (IK) "relational" systems thinking approach as holistic, illustrated as an integration of dimensions, with boundaries that are less defined, implying a dependency of understanding between dimensions. The concepts of "space" and "place" are key concepts in which systems thinking uniquely informs geospatially mediated solutions, reconceptualizations of the problems, and theoretical approaches. Note that knowledge of place is represented within the IK sphere. (See Section 2.3.). "Tools and Processes" occupies the space between these spheres. Cybercartography, described in Section 3.1, is a geospatial process and toolset that the authors have used to bridge these broad concepts and peoples of different knowledge systems. Cybercartography is presented as one example of a mediation process that can bring together "knowledge of place" and geospatial data models ("space"). Figure modified after Reyes et al.'s [23] Figure 3.1.

The next section introduces background concepts in systems thinking approaches and how these approaches uniquely inform geospatially mediated solutions. Then, we refer to sections of the paper to follow that describe how systems thinking serves to redefine challenges and accommodate shifts toward operationalizing the SDGs. The focus is a shift from top-down approaches toward local, community, and Indigenous leadership centered on holistic socio-cultural-ecological aspects for timely new avenues to build true resilience to climate change.

*Introduction of Systems Thinking Approaches*

Cybernetics was formulated as an aggregate of information and communication theories describing holistic organizational patterns and relational symmetry in social, natural, and man-made (e.g., machine) structures. Gregory Bateson [20]. was a polymath semiotician who blurred the lines between social and natural science disciplines to elucidate wholeness and interconnectedness in a meta-science approach termed an "ecological philosophy". Building on the works of Norbert Wiener, W. Ross Ashby, and Warren McCulloch, Bateson's cybernetic approach concerned the mind, including collective meaning-making and the symbols that encode meaning, and the emergent properties or outcomes that arise from cohesive interrelated, interconnected systems. In the context of this paper, a cybernetics perspective suggests the following:

1. Diversifying and decolonizing technology may include mapping and encoding meaning from diverse conceptualizations (if possible and ethical), and/or using "concept and meaning mapping" as a technological tool for engagement between knowledge systems;
2. Bottom-up, community-driven initiatives have an appreciable, aggregate effect toward significant macro-economic and global environmental goals.

A cybernetics analysis of global challenges like climate change, ecological collapse, and environmental injustice frames these issues as outcomes of complex, adaptive systems. These systems cannot pivot toward substantive change using problem-solving tools derived from those same systems (e.g., economy-based solutions within economic structures that remain dependent on exponential productivity and exploitation of nature and humans) (e.g., [2,20]) but systemic outcomes can change through creating positive adaptive feedback loops. Wiener described first-order cybernetics during the Cold War era, an era of rapid capitalism and consumerism at the beginning of the space race and the coming age of computation. First-order cybernetics is concerned with the feedback loops created in an attempt to regulate or control complex systems from an objective science and engineering approach [20]. These feedback loops inhibit inertial shifts toward changes that are necessary to balance or sustain natural systems when the measures of objective regulation and control become unsustainable. Second-order cybernetics offers a reflexive approach that appreciates the observer as part of the system itself, the cybernetics of "self-adaptive complex systems" [29,30]. This concept is described in "Design as Participation", "You're not stuck in traffic—you are traffic" [31]. In this approach, the observer appreciates that they are part of the phenomenon being observed. The reflexive approach is discussed in this paper, as it offers ways to respond to the significant scientific and dire socio-ecological challenges faced globally by asserting that participatory power lies in interconnected, complex, self-adaptive systems. The power to make systemic change is decentralized. In this way, local community initiatives toward sustainable human–nature interactions and climate change resiliency become as nodes in a decentralized, polycentric system that act both independently and interconnected through a direct or indirect focus toward similar outcomes. A cybernetics perspective suggests that positive feedback loops and durable and effective systems can develop but are dependent on diverse distributed nodes acting as multiple leverage points. From these resilient systems, coherent systemic change eventually occurs as an emergent outcome.

Realizing sustainable futures and polycentric systems require a myriad of data and information from diverse conceptualizations (i.e., knowledge systems), and geographic information systems have served as mediators for sharing knowledge across differences [19]. However, in a reflexive approach, "the role of the observer is appreciated and acknowledged rather than disguised, as had become traditional in western science" [30]. This approach calls for a critical self-analysis of the research methods themselves, wherein questions arise, including the following (as described by Pulsifer et al., 2005):

- Is it appropriate to encode and model all kinds of knowledge (e.g., IK)?
- How might this modeling affect knowledge access and control?
- What are the risks of appropriation, extraction, and misinterpretation?
- What are the risks in "diluting" or "reducing" relational knowledge, or other ways that might negatively transform it? [32]

Melanie Goodchild, Anishinaabe (Ojibway) systems thinking and complexity scholar, has critiqued Western systems thinking as having the same epistemological foundations as the analytical scientific method that, compared to "wisdom in action" of Indigenous "relational systems thinking", still suffers from "fragmentation and isolation" [33,34]. Goodchild goes on to say that to bridge the gap between worldviews, it is necessary for Western academics to see the privilege in *not having to* recognize other ways of knowing; furthermore, Goodchild writes that ideas coming from a different worldview have been entirely outside of the scope of Western conceptualizations (ways of thinking) and mindsets [33]. Bateson also recognized that Western systems were on an "epistemological runaway train" [19]. From a Western academic approach, this paper therefore discusses the reflexive approach

of second-order cybernetics as a work in allyship, seeking to further expand Western conceptualizations for shared understandings through discussing "complementary conceptual underpinnings" [8]. Although reductionism is the norm in many Western disciplines, Goodchild has described systems thinking as an "emergent system that melds the 'formal' and Indigenous knowledge systems" and wrote that "this type of cross-epistemic dialogue is now" [33–36].

Goodchild's reference to the "formal" knowledge system is the Western approach of formalizing knowledge (e.g., written), as opposed to Indigenous knowledge held culturally and often conveyed through oral tradition. This paper will discuss how formal Western knowledge has provided the structure for digital technologies, encoded as computer or "formal" logic. Essentially, digital infrastructures are based on standardized, conformant, top-down, and singular (i.e., colonial) conceptualizations of reality. In an increasingly technologically mediated world, particularly with generative AI playing a larger role in information sharing and decision making, there is a critical technological divide between dominant and non-dominant worldviews. This divide is in terms not only of accessing technology, but also of participating in its construct and how meaning is logically encoded. It has been suggested that this deepens global power and resource imbalances [37]. As such, there are movements to decolonize digital technologies (e.g., [38]). We lean on the "Indigenous Protocol and Artificial Intelligence Position Paper" [37] when asking how IK might be recognized as valuable to reorient technology to help inform processes (scientific, civic, policy, and otherwise) of decision making, and how might we collectively "imagine futures with technology that contributes to the flourishing of all humans and non-humans".

In Section 3 of this paper, we frame these concepts as couched around one implementation that the authors have used as a collaborative, cross-epistemic approach that leverages the systems components described in this paper (Figure 1), "Cybercartography" (e.g., [39–44]). Cybercartography is a community-based process that uses linguistics, semantics, and geospatial technologies as mediating platforms. This approach also includes a technological toolset uniquely suited to non-Western, narrative-based IK systems that integrates cultural, historical, linguistic, economic, and social data with geographic and cartographic information [45]. When deployed as a collaboration between IK holders, Indigenous communities, and Western researchers, the research-validated participatory methods are couched in ethical and just co-development principles. Deployments of these toolsets and frameworks are open source, which accommodates users to developing and deploying them independently. Again, in the framework of a reflexive approach, it is essential to recognize the potential of harms in delineating IK in the technological systems of modernity (e.g., [46]). Ethical frameworks, protocols, and further considerations are discussed in this paper.

A shift in operationalizing the SDGs from top-down approaches toward Indigenous-led action with a holistic socio-cultural-ecological focus may provide crucial and timely new avenues toward building true resilience to climate change. The Office of the High Commission for Human Rights (OHCHR) has indicated that the Climate Agenda and SDG frameworks currently lack cultural sensitivity and applicability to complex environmental problems specific to diverse local communities and Indigenous populations; with a recurring emphasis on GDP-focused growth, these agendas risk undermining Indigenous peoples' holistic development approaches and ecological sustainability practices [47]. These rights are asserted in the United Nations-adopted international Declaration on the Rights of Indigenous Peoples. Therefore, a further challenge of the SDG framework—as it includes guiding policies designed to support equitable climate change solutions for communities—is that without Indigenous peoples being provided a seat at the table, they will continue to be "left behind" again and again, and our systems may continue on their "runaway train" [19].

We suggest that taking a systems thinking approach, when non-Indigenous actors develop or co-develop methods and technology, may foster the use and development of intentional, ethical, and equitable digital technologies. Such technologies, implemented

by and with communities, are likely to be powerful tools used to meet global agendas for climate change and resilience by addressing issues at local and community levels. Several potential outcomes of this approach are discussed, including the following:

- Increasing participation in the processes and policies in place to meet climate change and environmental justice challenges from those holding diverse worldviews, providing novel insights and crucial understandings that are not currently presented to meet global challenges;
- Fostering a broader acknowledgment of, and adherence to, ethical frameworks and protocols for the development of digital technologies and use of geospatial data through a critical analysis of research methods and the asserted objectivity of the practice of science;
- Pathways to revolutionizing how Western scientists and technologists consider society, culture, and the environment as integral parts of moral and ethical checks and balances in the digital age;
- Diversifying AI and ML models for better collective information gathering and decision making, privileging knowledge systems with a basis in moral and ethical obligations toward life—both natural and artificial systems—to help spur revolutionary shifts toward global sustainability goals;
- Supporting the development of sustainable and resilient polycentric techno-socio-ecological systems through a myriad of data and information from diverse knowledge systems, with geographic information systems serving as a mediator for sharing knowledge across differences.

## 2. Semantics, Cybernetics, and the 'Knowledge in Place' Challenge of Big Earth Data

There is a strong movement underway to call attention to the stagnation progress on the SDGs due to a lack of frameworks for monitoring and assessing them. The United Nations Global Geospatial Information Management Committee of Experts (UN-GGIM) has been tasked with employing the power of geospatial data and scientific tools to navigate complex relationships among social, environmental, and economic objectives toward the UN Climate Agenda. It is said that everything happens somewhere—assessing environmental, social, and statistical data through geospatial data, tools, and technologies provides immediate observations for crisis mitigation and long-term adaptation. Geospatially related technologies have proven indispensable tools used to effectively identify and track ecological and humanitarian crises, thus assisting in rapid responses to dynamic relief and justice efforts on the ground [48]. Even though there have been decades of consistent remote sensing observations and climate and environment modeling and the interplay with society, gaps remain in the capacity for geospatial technologies to provide a deeper, more comprehensive understanding of the complex and multi-factorial challenges inherent in the challenges described in the 2030 Climate Agenda. It has been suggested that a fundamental gap that is collectively faced is between data and information about "space" and knowledge about "place". In this context, we discuss "knowledge about place" as the rich interrelation (holistic) knowledge from direct experience of human–ecological interactions over millennia that commonly describes IK [8–10].

The discussion in the following sections on geospatial and related digital technologies ("GeoSemantics") articulates "knowledge in Place" and the under-representation of diverse worldviews in technology development. The synthesis of current understanding from these fields provides caution in the development of technologies (such as AI): *Who* is mapping the "meaning", and *what realities*, or worldviews, are being hard-coded into the structures of digital technologies?

### 2.1. Semantics and Geocybernetics

One challenge in producing Big Earth Data and GeoAI that accommodates the specificity of "knowledge in place" is the semantic heterogeneity of multiple sources of knowledge, data types, and forms of geospatial data. This heterogeneity is not properly addressed

through metadata standards alone. There is an important distinction between syntax (e.g., metadata standards) and semantics that helps elucidate that the production of data and information is not the same as the production of actionable knowledge. Computer technologies do not operate with an understanding of the physical world, but on a formal syntax, a manipulation of symbols. The relationship of symbols in syntax to causal reality is not necessary in modern technological computations. In computer science, the relationship between symbolic coded representations and concepts (and their relational meaning) in the real world is "semantics" [49]. Semantics, as a general term, implies the meaning or cognitive structure of shared understanding. In sub-branches of various fields including linguistics, philosophy, logic, and computer science, semantics is the study of referential and contextual meaning. Semantics assigns meaning in digital technology, and thus, an avenue through which we might assert ethical meaning (which extends to decision-making capacity) in its design.

In computer science, semantic models take symbolic representations of knowledge, concepts, and reasoning that are easily (and sometimes commonly) understood by humans. One utilization of semantic models is to make relational meaning and interpretations explicit and precise for machine computation [49]. (As discussed in Section 2.3, another utilization of semantics is in collaborative concepts and "meaning" mapping as a technological tool for engagement between knowledge systems.) Shared vocabularies are currently being used across the internet, with over 10,000 websites using schema.org semantic mark-up. Founded by Google, Microsoft, Yahoo, and Yandex, and continually developed by an open community process, schema.org vocabularies enable more than simply human-readable website to be made available but serve to automate content processing through self-described websites, by, for example, providing intelligent and contextualized information.

As geospatial technologies commonly use ML models, it is worth contrasting semantic modeling with AI and ML models for clarity. AI and ML models seek generalization and implied meaning, and semantic modeling seeks granularity and specific meaning [49]. For example, data analytics-based AI and ML models could discern that a forest is composed of a set a tree species from a pre-specified list of species and associated observation variables but, unlike semantic models, could not track the fact that the fate of chimpanzees in the forests of Gombe depend on the flesh of native palm oil trees, or that protecting both is required to preserve both biodiversity and local ethnobotanical heritage and human needs. ML predictive data models work as the core of inference, predictions, and meaning that a system is capable of deriving. In building on traditional ML-based computation, the use of formalized semantic vocabularies shared across the web can allow the fusion of multi-source heterogeneous geospatial and temporal data, and to some extent, ameliorate interoperability problems. The methods, standards, and architectures of formalized semantics vocabularies can then allow AI to make use of "smart" data, with systems capable of analyzing text and patterns (e.g., Large Language Models (LLMs)) to autonomously mediate how geospatial and associated data are represented.

As opposed to the big data focus of these models, Geocybernetics, although a relatively new field of research, has focused much of its foundational development on the scale of community data and knowledge. López-Caloca et al. [24] formalized a semantically driven model for Geocybernetics that relied on a metasynthesis of stories, such as those told by local communities (see the Cybercartography Atlases described in Section 3.1). In this approach, key concepts from storytelling serve as the backbone of "chunk" semantic concepts in a novel knowledge management and integration model. The "chunk" concepts are adopted as formulas related to the societal processes of the community stories told, and knowledge networks of meaning emerge from the formalization of the heuristics, social agreements, and other defining details of the culture [25]. Therefore, "qualitative prose becomes a key component" in the process of Geocybernetics, and its semantic mapping builds "transdisciplinary bridges (implicit and explicit) that connect different knowledge domains" [24,50]. Communication between semantic knowledge domains

(including dialogues between people of different knowledge systems) is necessarily an inclusive process, creating a "transverse axis on which actors travel as they provide their explicit knowledge frameworks for the evolution of ideas and actions that respond to societal problems." [23].

*2.2. Systems Thinking Theory and Background of "Knowledge in Place" Representation*

The discussion of semantics, formalized semantic vocabularies shared across the web, and Geocybernetics is to provide background for a caution of AI and to ask guiding questions for systems and model development, geospatial and otherwise: *Who* is mapping the "meaning", and *what realities*, or worldviews, are being hard-coded into the structures of digital technologies? Will it be possible to build better models that help to meet the challenges described in the SDGs, if they are co-developed with peoples that hold "knowledge about place"? Knowledge about place includes the rich interrelation (holistic) knowledge from direct experience with human–ecological interactions over millennia [8,10,11]. While we draw on Indigenous scholarship in this work, we recognize the dominant coloniality of technological and academic institutions. We recognize that this has broad implications for the kinds of engagement, participation, and governance accommodated in technologically mediated spheres.

Geospatial models offer predictions that are only as accurate and as contextualized as the training data they are built from; similarly, the "knowledge" and inferences that are made through AI are representative only of the meaning that is mapped out in its formal logic. This is a central concern and an inherent risk in using standardized vocabularies; they do not represent the diversity of conceptualizations of peoples in the world. Standardized vocabularies are the most scientifically useful and interoperable, as they seek to deconstruct knowledge into highly formalized, minimalist categorizations, essentially dividing reality into subcategories of time and space (e.g., [50]). Natural language constructs—human interpretations of real-world phenomena, or "reality"—are represented using first-order logic (FOL), a suite of logical formalizations "sufficient to achieve a good approximation" that consequently "sacrifices a considerable part of the semantics [or, meaning] in order to achieve computability" [51]. A semantic characterization of space, time, and spatial and temporal concepts also requires this level of formalization [51], and it has been argued that it suffers from the same limitations [52]. This reductionist approach further formalizes meaning, in the technological world, into one single dominant worldview world. It has been argued "that universality through [geospatial]. ontologies can potentially perpetuate homogenization of concepts, thus contributing to assimilation of Indigenous peoples" [52]. The 2022 Montreal Declaration [53] provided guidelines on the development of AI to be of service to human well-being and with strong democratic legitimacy, including the 'Equity Principle' and the 'Diversity Inclusion Principle'. These principles state that AI development and use must "not lead to the homogenization of society"; "take into consideration the multitude of expressions of social and cultural diversity present"; and "be designed and trained so as not to create, reinforce, or reproduce discrimination" [54]. In 2023, the Future of Life Institute published an open letter, signed by over 13,000 signatories including AI developers, experts, and ethicists calling for a complete moratorium on AI development until serious mitigation strategies for the "societal scale" and "extinction" level risks posed by unregulated AI [16,53]. The complex landscape of AI, with its quickly evolving capacities and debates, is beyond the scope of this paper. However, in consideration of the serious concerns expressed by leaders in technology about the cultural diversity and lack of ethical constraints in AI, we reflect on a critique offered by Melanie Goodchild, Anishinaabe (Ojibway) systems thinking and complexity scholar—the way we seek solutions in our globalized world share the same epistemological foundations as the problems that created them [19,33,34,36].

Semantically encoding "reality" from one dominant worldview means that solutions that may arise from an increasingly generative, automated technological system will leave out ideas coming from different worldviews [53]. Conceptualizations and solutions arising

from Indigenous conceptualizations are entirely outside of the scope of Western conceptualizations and mindsets (e.g., [33]), yet may provide key insights for meeting global challenges, particularly informing how humans relate to their environments. Goodchild compares Western approaches to the "wisdom in action" of Indigenous relational systems thinking [33]. An extensive body of scholarly work has emerged in interdisciplinary studies of philosophy, science and technology, geography, and anthropology that offers an interrogation of the ways that dominant discourse, practices, institutions, and technologies shape the worlds in which people live and the solutions that are possible, or confined, within it. Known as "post-structural", these investigations address semantics as essential for reimagining sustainability and environmental governance "because consolidated assumptions regarding the nature of categories of being in the world shape human action in the world, and thus have ethical, including ecoethical, effects" [55]. Interrogating the semantics used in geospatial and other digital technologies (e.g., the "tools and processes" space in Figure 1) through an Indigenous relational systems thinking lens (e.g., holistic approaches, as illustrated in Figure 1) may offer the opportunity to reframe how the most pressing current socio-environmental challenges and solutions are formulated.

There are movements to reorient the semantic development of AI frameworks that centralize Indigenous epistemologies and "integrate [digital technologies] into existing ways of life, support the flourishing of future generations, and are optimized for abundance rather than scarcity" [56]. Jason Edward Lewis, lead author of the 'Indigenous Protocol and Artificial Intelligence Position Paper' [37], writes about reimaging AI based on whole-systems understandings that remove anthropocentrism and "extend intelligences" to that of non-human "kin". These kin include digital technologies and AI, the non-human biological world, and the geological and ecological environs that create the container for all of life [57]. This vision of AI can be thought of as a technological dimension of the Indigenous relational systems theory. Such a reconceptualization is suggested to be an avenue toward reorienting digital technologies that consider respect and reciprocity of all kin connections—centralizing respect of life and natural systems as encoded in technology. This reframes AI from being a "tool" to a kin connection, and by this nature, AI is developed with care and intention. Lewis describes "prioritizing human flourishing" [37], referencing cyberneticist, mathematician, and philosopher Norbert Wiener, and goes on to describe that decentralizing humans in this conceptualization puts humans in the same moral and ethical framing as all other kin relations, and as such, can neither be objectified nor "used as mere tools" [20,57,58]. Expanded notions of the agency of nonhuman actors, as subjects rather than objects, is a central and key contrast between Western and Indigenous conceptualizations [56]. Engaging with or integrating the expanded semantics of non-humans as subjects, with inherent agency and morality, rather than mere tools or resources to be used for human services, requires deep questioning about both ecology and technology, and our relationships with and obligations to them [58,59]. Humans and non-humans being on an equal moral and ethical plane entails a consideration of respectful relations, and thus potentially reframing the root of global challenges in consideration of the well-being of all human and non-human agents.

We acknowledge Goodchild's critiques and (leaning on Lewis' approach) offer two points of comparison from Western cybernetics as a potential bridge between different worldviews. First, Weiner describes Goodchild's critiques of insufficient solutions arising from the systems that created them as 'feedback loops'. These feedback loops inhibit inertial shifts toward changes that are necessary to balance or sustain natural systems when the measures of objective regulation and control become unsustainable [20]. Second, cyberneticist Gregory Bateson described the insufficient solutions as consequent of an 'escalating run-away conformity' of digital technology:

> *"Out of our ignorance of a total system of relations and their complex functioning—and out of the selection of "individuals" (or peoples, or countries) as independent, isolatable things—we can fall into pathological patterns. We can get ourselves into "double-binds," where destructive behaviors are reinforced by conscious efforts to mitigate them. In*

*double-binds the message, at one level, is contradicted at a different level, and pushing
the message inadvertently reinforces the pathological behavior"* [19].

=Manifestations of modernity's systemic "pathological behavior" include the climate
crisis, the sixth mass extinction event, and the inability to adequately meet local and global
sustainable and justice goals. One antidote may be found in radically different, holistic
conceptualizations of the relationships between humans and the natural world., Indigenous
technologists that are centralizing respect of life and natural systems as encoded into digital
technologies (e.g., [37]) may help the modern world to step off the socio-technological run-
away train. As AI becomes ever increasingly used for collective information gathering and
decision making, diversifying knowledges at its base and privileging knowledges that hold
the moral and ethical obligations toward life—both natural and artificial systems—may
help spur revolutionary shifts. Bateson argued that wrongly siloing these interconnected
aspects of human experience—removing social and environmental morality and ethics from
science, engineering, and technological pursuits—creates an impossible dilemma, omitting
humanity from its natural embeddedness in all these aspects. In short, Bateson predicted
that seeking to control natural environments to address problems without moral and ethical
checks and balances would hasten ecological catastrophe and ultimately disempower and
cause decision paralysis [19]. A fundamental flaw in modern sustainable development
practices is "us against them" politics, positioning humans against nature, devoid of
"ecological consciousness" in which humans are innately embedded [19,59].

Ecological consciousness is inherent in holistic Indigenous conceptualizations of
"space" and "place". Sheridan and Longboat (2010) [60] describe this as follows:

*"Conceiving of imagination without sourcing its ecological origin contributes to and ex-
tends anthropocentrism consistent with minds unwilling to naturalize to their surroundings. . .*

*Minding all things performs the spiritual conservation of all things. All things comprise
the Indigenous mind and Indigenous minds are composed of all things."*

Expanding formal vocabularies and semantic models in digital technologies to attend
to an "ecological consciousness" is a complex challenge. As discussed in following sections,
we acknowledge that codifying and modeling is not always appropriate, or possible, for
all forms of knowledge, and comes with potentially serious risks, some of which that
may repeat historical harms. While acknowledging these risks, to forward inclusive social
transformations and meet the needs of all humans, Indigenous and non-Indigenous, in the
digital age, engagement with Indigenous peoples and people in communities is paramount.
Section 3.1 describes use cases of Western researchers working with IK holders to integrate
'knowledge of place' and traditional culture into digital technologies in a container of
co-development for equitable technologies that are based on, and explicitly support, IK.

Systems scientist Helene Finidori studies how worldviews and practices co-evolve
to advance human thinking and produce new emergent worlds of social transformation.
This transformation is possible when sense-making can be realized and shared, as with
co-developed digital technologies with semantic meaning. Finidori [2] has described the
potential for this emergence as durable and effective systems that depend on a variety
of distributed nodes, acting as multiple distributed leverage points, to provide a rich
and resilient polycentric basis. From these resilient systems, coherent systemic change
eventually occurs as an emergent outcome. For the challenges presented in the SDGs, in
this cybernetics perspective, the much-needed systemic change would arise from aggregate
agency—each of the distributed nodes in the system having agency and autonomy.

A systems thinking and reflexive approach to semantic web and geospatial computing
models asks *who* is mapping *what* into increasingly automated information technologies.
In this approach, the authors are careful not to presuppose that purely rational, top-down
approaches are sufficient, or that standardized, singular understandings of meaning serve
to build a knowledge infrastructure that is agile enough to action science amid complex
socio-ecological collapse. Instead, we suggest that systems thinking frameworks may serve
to revolutionize the way Western scientists and technologists consider society, culture,

and the environment as integral parts of moral and ethical checks and balances in the digital age.

In the next section, we discuss the challenges and opportunities presented in using geospatial technology as a mediator between knowledge systems, bringing together Indigenous land-based "knowledge about place" and the geospatial models of "space" data and information (see Figure 1). Section 3 discusses further interventions and cross-epistemological co-production models that the authors have used. Section 3.3 describes risks, implications, and potential harms in codifying relational IK into technologies to include the reduction in meaning if extracted from its knowledge base, culture, or contextual meaning, as well as the caution in employing such methods; the historical, negative consequences of modernization must be acknowledged.

### 2.3. GeoSemantics: Space, Place, and Belonging

The concept of "space" is of concern to geomatics technologists, as it details the observable or measurable features of location, often derived from remote sensing, with its abstractions stored and processed by a machine; "place" concerns the meaning that humans invest into locations, and how humans understand the world through that located meaning [61]. IK systems are recognized as holding complex understandings about place: practicable and sustainable human-ecological interaction that is embedded in the cultural knowledge of reciprocal relationships, developed over time. This knowledge of place is an example of Systemic Literacy [2], which takes into account how systems function, beyond "mechanical" aspects and includes their political, philosophical, psychological, emotional, existential, relational, anthropological, and epistemological dimensions. Finidori [2] describes resilient, agile socio-technological systems from a cybernetics perspective as those that develop and integrate relational intelligence and systemic consciousness; this conceptualization is central to Indigenous sense of place [2].

Carolyn Briggs (Boon Wurrung Elder in Residence at RMIT University) wrote in her 2020 paper "Bridging the geospatial gap: Data about space and indigenous knowledge of place" [61] that Indigenous knowledge of place is not well-served by current digital geospatial technologies. Three key challenges to meet this gap were identified as follows:

- The overrepresentation of digital data about space, rather than knowledge of place;
- A lack of facility in differentiating access to knowledge and enabling Indigenous data sovereignty;
- A lack of facility in supporting and sustaining relationships between Indigenous and non-Indigenous peoples [61].

Although these challenges remain, Briggs et al. suggest that identifying technological opportunities could offer a pragmatic pathway to more rapidly bootstrap new approaches beyond simply technological "fixes". New approaches are necessary in addition to technological innovations that may result in local constructive responses but do not adequately move toward the systemic change that is necessary [61]. As described by Goodchild and Bateson, Briggs et al. [61] asserted that an attempt to meet challenges with more and more technological solutions risks deepening problems from a lack of consideration of the total system of relations and their complex functioning. Integrating collaborative knowledge, co-developed interventions, and participatory methods may enable the collectively sense-making of the dynamics at play in our socio-technological and socio-ecological systems and integration of their different dimensions, to enable novel systemic interventions.

Such novel interventions may be described as by Finidori [2] in the context of a geospatial technological system, involving a set of mediating capabilities and tools to perform the following:

- Make sense of patterns and growing volumes of information and knowledge;
- Leverage agency and the complementarity of perspectives, knowledges, and capacities to include Western and Indigenous sciences;
- Help change agents—change communities to bring knowledge of environment and place to where they are located—to contribute to the evolving knowledge of the whole;

- Realize a decentralized nodal network approach (e.g., machine-referential semantic web vocabularies) with multiple distributed leverage points that may form a coherent systemic change as an emergent outcome of aggregate agency [2].

Section 2.1 of this paper discussed one potential utilization of semantic modeling in terms of formalizing the relational meaning and interpretations of IK more explicitly and precisely for machine computation. One other utilization of semantics is collaborative concept mapping as a technological tool for Indigenous knowledge representation and engagement between those of different knowledge systems. This approach was taken by Pulsifer et al. [62] in a co-developed semantic model of Inuit knowledge. This model maps a foundational understanding of Inuit knowledge of caribou and other key subsistence species in Nunatsiavut, one of the four Inuit regions of Canada. The Nunatsiavimmiut knowledge map was digitally linked to text, multimedia, and GIS land and resource planning maps that were already in use in Nunatsiavut. In this project, the Inuit knowledge holders who worked with Pulsifer et al. used food security, wildlife stewardship, and GeoSemantic concepts as a starting point for Inuit knowledge to inform local decision makers, multiple generations within the communities, and Western scientists [62]. GeoSemantics, with holistic systems thinking approaches, may provide potentially powerful tools for meeting the gap between spatial information and knowledge about place, with the power to provide more holistic perspectives from the context of interconnected conceptualizations of human–nature interaction that is symbiotic and sustainable.

Iliadis et al. [63]. described another approach of applying the semantic formalization of polar data with Arctic Indigenous communities. As exemplified in this study, IK systems were recognized as holding complex understandings of human–ecological interaction— practicable, reciprocal relationships developed over time as a cultural approach that predate modern data acquisition and have not been considered in reductionist approaches. Moreso than its practicality and importance in the "enhanced cross-cultural understanding of the environment and improved resource management" [63], the inclusivity of these communities is at the heart of geoethics. Iliadis et al. [63] goes on to say,

> "…if researchers and residents ignore this increasingly dominant form of knowledge representation, their voices may be silenced in key knowledge construction and information policy-making processes."

In the next section, we discuss an applied geomatics model that the authors have explored to bring together the complex ideas presented in this paper to grounded, actionable work between peoples of different worldviews. The model is a socio-technological approach, and one that seeks just and equitable knowledge co-production, using geospatial tools as mediating technologies [32].

## 3. Discussion

### 3.1. Applied Geomatics and Knowledge Co-Production

Cybercartography has developed as both a community-based process and digital technology tool stack (e.g., see "tools and processes" in Figure 1) that the authors have applied as cross-epistemological interventions with communities. Some of these Cybercartography projects include the following: the Clyde River Knowledge Atlas (shown in Figure 2) [64]; the Residential Schools Land Memory Atlas [65]; the Lake Huron Treaty Atlas [66]; a Collaborative visual repatriation project with Inuit in Nunavut [67]; and the Inuit Sea Ice Use and Occupancy Project [68,69], which involved community youth in environmental conservation while connecting to their elders and conserving culture.

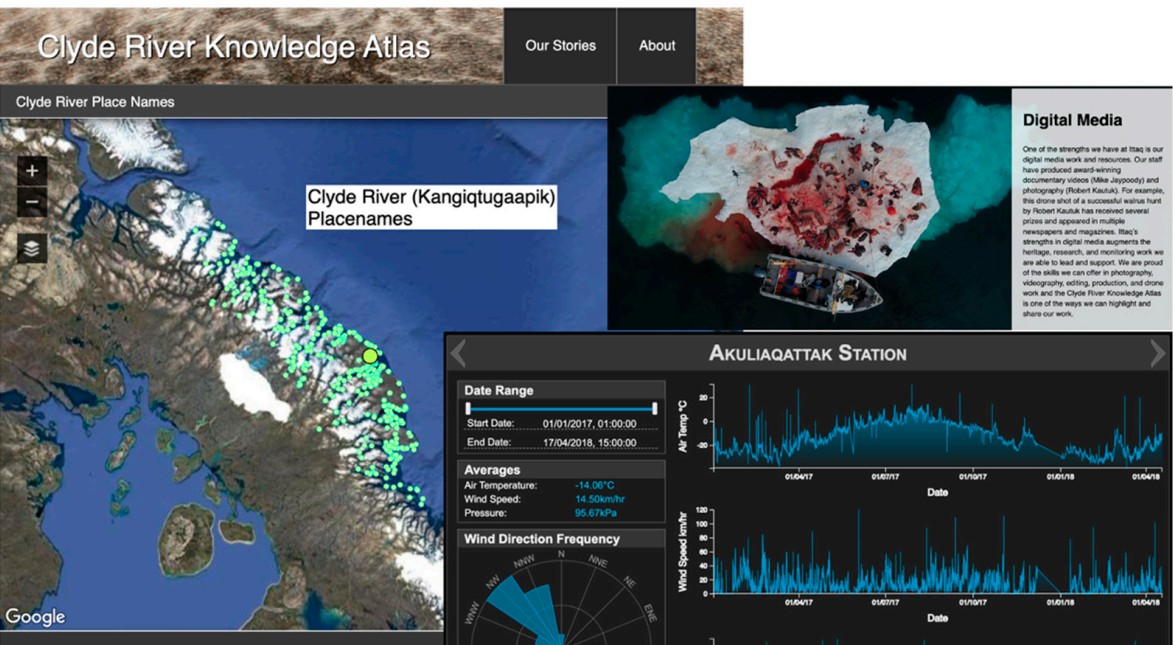

**Figure 2.** Example of one Cybercartographic project. Images from the Clyde River Knowledge Atlas [64], co-produced by the Ittaq Heritage and Research Centre [70] and the Kangiqtugaapik community. Insets shown include the following: (**left**) *Clyde River Traditional Placenames* map, the names themselves holding knowledge and culture; (**top right**) *Digital Media*, showcasing award-winning community documentaries and photography; (**bottom right**) *Clyde River Weather Station Visualizer*. The atlas provides a digital location to protect and share traditional Inuit knowledge. The atlas also houses data and research from non-Indigenous researchers, made available for the benefit of the people and lands, in accordance with the Inuit of the Circumpolar (ICC) *Protocols for Equitable and Ethical Engagement* (e.g., "Nothing About Us, Without Us"; see Section 3.3).

Cybercartography incorporates linguistics, semantics, and geospatial technologies as mediating platforms (as described in the previous section). This technological toolset is uniquely suited to non-Western, narrative-based IK systems as it integrates cultural, historical, linguistic, economic, and social data with geographic and cartographic information [45]. When deployed as a collaboration between IK holders, Indigenous communities, and Western researchers, the research-validated participatory methods are couched in ethical and just co-development principles. Deployments of these toolsets and frameworks are open source, which accommodate users with developing and deploying them independently.

A multitude of case studies and successful implementations of Indigenous-led participatory mapping approaches are described in the 2022 "International Journal for Geo-Information (IJGI) Special Issue on Mapping Indigenous Knowledge in the Digital Age" [71]. One study describes a stepwise process of a participatory workshop development and implementation by Andrade-Sanchez and Eaton-Gonza'lez [72] in working with the Kumeyaay Peoples of Baja California. Workshops of this type are intended to build a team that includes a technical group and the community in a dynamic exchange of knowledge that continues throughout the project. First, the participants develop a community self-diagnosis about natural resource management, conservation, and climate resiliency efforts. Second, they identify the main objectives and a community strategic plan to then address the problems. The community is supported in this process by scientists and researchers in the use of cartographic and remote sensing tools and technological platforms. In other projects, the scientists and community co-develop Cybercartography mapping approaches (e.g., [40–44]) to meet complex community-specific environmental problems; Indigenous leadership is available for critical areas at the intersection of culture and climate change vulnerability. The participatory processes developed in Kumeyaay

communities involved the mapping of plant species distribution and the assessments of their conservation status, which supported community training, the mapping of forest pests, and the construction of a Cybercartographic atlas for natural resource management decision making.

As is key in many of these studies, the local knowledge yielded high-quality data given the holistic mapping of the relationships among environmental variables and knowledge of their local and broad implications. This results in a deeper understanding of the data and actionable knowledge for resilience or mitigation strategies. This understanding incorporates several dimensions into one analysis, including indirect cultural provisions, biodiversity indicators, and local economic stability and resilience. Intrinsic values (as described in the Intergovernmental Science-Policy Platform on Biodiversity and Ecosystem Services (IPBES) Report [73]) are indicators that are not well delineated through Western scientific methods. Nonetheless, when these holistic understandings from "knowledge of place" can be incorporated into one process of analysis, they become powerful tools for the management of natural areas, local decision making, and adding rich dimensions to solve problems as identified by local communities.

Tools that integrate narrative-based IK (including intrinsic or spiritual values as well as cultural, historical, linguistic, economic, and social data) with geographic, remote sensing, and cartographic information work to bridge the digital divide and enhance participation in a global effort for climate change mitigation and resiliency. These approaches can thus be an avenue toward meeting the challenges posed by the SDGs, bringing radically different understandings of human–nature relationships and enabling local communities around the world to use geotechnologies for identifying environmental problems and enacting practical solutions. These collaborations build communication channels, bridge data gaps, and enhance remote learning facilitation for Indigenous participants.

*3.2. Mediation for a Shift in Worldview*

Collaborative work in spaces of knowledge co-production or knowledge exchange with IK holders often afford Western scientists, researchers, and technologists a radically different perspective. Embodied experience with the land is a key means of fostering deep relationships over time, with the inseparability of people and place a common theme in Indigenous worldviews [18,74,75]. This inseparability is beyond Western conceptualizations of the human/nature dichotomy to the embodied experiential knowing of sociality and kinship with natural environs [18]. A common phenomenon among Western researchers who begin to exchange knowledge on land with IK holders can be likened to experiencing an "Overview Effect", enabling a more expansive perspective of interconnectedness. First coined by Frank White in 1987, the Overview Effect is a cognitive shift experienced by astronauts who, upon viewing Earth from an overview perspective, adopt an acute understanding of the precious and interconnected nature of life on Earth and the responsibility to care for it. In developing relationships with peoples whose relationship with the land is more than an essential part of their survival but is intrinsic to their identities [61], scientists and researchers without this connection may begin to develop some relationship with the land and thus develop a capacity to appreciate a wider relationality with the natural environment. For geospatial scientists, this relationship-building may afford a mediation of the missing "knowledge of place". More broadly, this relationship-building may mediate a shift in Western researchers' worldviews to allow for, in Bateson's words, a new "set point" [19]. This new "set point" for non-Indigenous people include personal yet crucial insights and deep, embodied understandings of non-colonial worldviews as equally salient to meet shared, global challenges. Embodied understandings do not arise from an intellectual exercise but emerge through engagement.

In the course of research, it is not always possible for researchers to physically visit the land they study, and neither are IK holders always available or amenable to share their knowledge on their lands in some contexts. In the absence of gaining perspective on the land with IK holders, one potential experiential pathway is the type of Cybercartographic

atlases described in this section. As previously discussed, Cybercartographic atlases and similar digital tools may enable greater digital representation and communication of Indigenous worldviews and accommodate greater IK visibility in key national and international policy-making fora. In this section, we further suggest that these tools may provide a digital platform for Western researchers to expand perspectives, learn more about relational worldviews, and strengthen allyship in science and technology.

Previous sections of this paper have described validated research methodologies and successful implementation use cases of knowledge co-production or knowledge exchange with IK holders to diversity cartographic and geospatial science and technology. The next section describes existing mechanisms, guiding protocols, and principles for ethical engagement with Indigenous groups (particularly in the Canadian context), data sharing and data sovereignty, and technological co-development (to include the ongoing development of AI-mediated technological frameworks). We further advocate for such protocols to be referenced and made explicit in science–policy guidance documents prepared by UN-GGIM committees that seek to advance action on the SDGs and the Climate Agenda. Acknowledgement of and adherence to these principles would support and strengthen science–community–policy interfaces on a basis of equity and increased diversity.

### 3.3. Guiding Principles and Cautions for Geoethics

Even constructive responses resulting from technology and innovation, empowering scientific and local communities, may have broader implications to include the global exposure and broad dissemination of Indigenous ontologies [66]. This may result in extraction, loss of ownership, and continuing knowledge colonialism. As non-Indigenous academics, exploring the implications of these risks is an ongoing exercise in self-reflection, honesty, and humility, DePuy et al. (2022) asks *"What does outsider advocacy for a plural reconceptualization look like, while avoiding romanticization or the continued privileging of outsider perspectives? And, "is it possible to contribute without falling into the trap of reinforcing western-based institutional knowledge at the expense of marginalized and subjugated ontologies?"* [56]. We hope that staying with the discomfort of embracing pluralisms in science and technology frameworks may offer allyship and an ethical approach to technological co-development. As a matter of academic ethics and moral responsibility, and in a conscious attempt to minimize potential risks, we work to continually recognize historical and ongoing negative consequences of modernity, from which digital technologies are not easily separated. Furthermore, Iliadis et al. [63] cautions against the implications of codifying and analyzing Indigenous knowledges using a logical, reductionist framework that is driven by Western scientific ideals; there is a destructive potential of reducing meaning into a form that can be readily extracted from its own knowledge base, which may serve to break down the culture and its contextual meaning [63]. These risks must be acknowledged. However, bridging the technical divide (and perhaps, bridging a chasm of understanding that separates peoples of different worldviews) cannot be ignored if it facilitates providing Indigenous peoples a seat at the table in global and national decision-making processes. Furthermore, we recognize IK as valuable to reorienting or remapping technology, and through positioning Indigenous leadership in these fields, to contributing to collectively imagined "futures with technology that contributes to the flourishing of all humans and non-humans" [37].

A mix of great progress and many challenges remain in the development of tools used to support ethical, self-determination in data sharing, and policies and other mechanisms related to Indigenous data sovereignty. In the Canadian context, local Indigenous and First Nations communities' rights have been codified by protocols that elucidate on their participation and engagement with, and contributions to, nation-to-nation and international fora, exercising their sovereignty in these spaces. These protocols and principles include but are not limited to CARE Principles, OCAP®, and ICC Protocols for Equitable and Ethical Engagement. These guiding protocols are described in this paper to provide a greater awareness of the protocols for equitable and ethical engagement, and they are not meant

to replace or dimmish existing engagement and data protocols that may already exist for individual First Nations communities.

The CARE Principles for Indigenous Data Governance were developed in consultation with Indigenous peoples, scholars, non-profit organizations, and governments, as part of the CODATA Data Ethics working group (Committee on Data of the International Science Council (ISC)). The CODATA group addresses the implementation of, and challenges that arise from, the UNESCO Recommendation on Open Science [76]. It arose from concerns about the secondary use of data and limited opportunities for benefit sharing. These principles are Collective Benefit, Authority to Control, Responsibility, and Ethics, and they complement the existing data-centric approach of the FAIR (Findable, Accessible, Interoperable, Reusable) Guiding Principles for scientific data management and stewardship. It is important to note that these guiding principles were established from the tension that Indigenous communities indicated to be present between (1) protecting Indigenous rights and traditional knowledges and (2) supporting open data, machine learning, broad data sharing, and big data initiatives.

In 2023, the CODATA working group published a Draft Policy Brief to meet the ethical challenges raised by the growing application of big data and AI, providing a basic consensus document on data ethics principles while championing global open data exchange [77]. One of the four themes in the draft document is "Ethics and Indigenous Data Governance". This theme outlines that international recommendations on Indigenous data governance are not "primary documents", but that self-determination, data ethics, sovereignty, and stewardship are to be defined by any Indigenous population, individually, and on their own terms. The authors note that the UNGGIM has co-developed, with the World Bank, the Integrated Geospatial Information Framework (IGIF), which are guidance documents that address the specific challenges associated with the management, use, and exchange of geospatial data and national strategies for the use of geospatial data to meet the 2030 Agenda and SDGs. Merodio Gomez et al. (2022) [78] explicated the potential misuse and monitoring concerns in the context of the Americas, with recommendations on ethical limitations (e.g., the Locus Charter) to ensure a balance between the risks and benefits of its use in South and Latin American countries.

Again, in the Canadian context, the First Nations Information Governance Centre (FNIGC) developed the First Nations Principles of OCAP (Ownership; Control; Access; Possession). These principles assert that Indigenous communities alone have control over data collection processes in their communities, and that they own and control how this information can be stored, interpreted, used, and shared. The FNIGC provides education and training on the First Nations principles of OCAP®, and it is recommended that these courses be taken by Western scientists, working directly with communities or not, to gain basic understandings of information governance and data sovereignty. As enshrined in the UN Declaration on the Rights of Indigenous Peoples (UNDRIP), essential to Indigenous peoples' rights are the self-determination of their economic, political, social, and cultural development; these enshrined rights go beyond economic and cultural development; this includes real-time knowledge and data in all its forms. It is within the realm of the self-determination of Indigenous peoples to have control over naming problems and solutions, as well as over the narrative, place names, symbols, and data and their use. OCAP training is available from their website and is recommended for ensuring ethical practices in science and technology.

The Inuit of the Circumpolar (ICC) is a registered NGO with consultative status with numerous UN specialized agencies and bodies, and it has developed the ICC Protocols for equitable and ethical engagement. Although written from an Inuit perspective, these protocols have wider applicability, as do the CARE, FAIR, and OCAP principles. The eight ICC Protocols are as follows:

1. Nothing About us Without us;
2. Recognize Indigenous Knowledge in its Own Right;
3. Practice Good Governance;

4.   Communicate with Intent;
5.   Exercise Accountability—Building Trust;
6.   Build Meaningful Partnerships;
7.   Information and Data Snaring, Ownership, and Permissions;
8.   Equitably Fund Inuit Representation and Knowledge.

A significant amount of work remains to ensure that policies, licenses, and data and digital technologies can be effectively implemented in the context of the principles and protocols discussed in this section. Many of the current data-sharing, technological co-development, and AI-mediated technological frameworks require extensive human interaction for current implementation and future development; we therefore offer a caution in this paper with respect to *who* is mapping 'meaning' and to *which* worldviews, principles, and protocols are being encoded into technologies as the next generation of computing evolves.

## 4. Conclusions

This study presents a review of cybernetics in the context of digital technologies, applied semantics, and ethics as a framework to approach the most pressing challenges faced globally, as described in the UN SDGs. With a synthesis of these interrelated spheres, the authors argue that cybernetics provides a Western systems thinking framework as a revolutionary way for scientists and technologists to consider moral and ethical checks and balances in the digital age. Cybernetics may also serve as a bridge of understanding to Indigenous Relational systems thinking—a framework that describes holistic Indigenous ways of knowing with inherent moral and ethical dimensions [33]. Indigenous relational systems thinking [33] offers Systemic Literacy [2], or deep interconnected understandings of complex dimensions, including agency of both humans and non-humans, and the interwoven complexities of the ecological, political, economic, philosophical, emotional, spiritual, and epistemological dimensions. Using a bridging approach, Western researchers might engage with or integrate expanded semantics of Indigenous relational thinking that considers non-humans as subjects, with inherent agency and morality. The consideration of 'non-human' also applies to digital technologies as 'agents'. A pluralistic approach to digital technology development—one that includes a diversity of knowledge systems—would require deep questioning about both ecology and technology, and our relationships with, and obligations to them [56,57]. This implies a consideration of respectful relations between humans and the ecological world, thus potentially reframing the root of global challenges in consideration of the well-being of all human and non-human agents. The systems thinking approaches reviewed in this study suggest that using holistic, relational lenses could radically shift the way we articulate and approach wicked problems as described in the SDGs. This would allow solutions to arise from diverse understandings, not alone guided by the data of Western science, but by values and moral or spiritual systems that guide humans relationships to nature.

This review study argues that through the relational activities of sharing across knowledge systems, (1) agile frameworks can be built and optimized for future-resilient sociotechnological networks capable of meeting local and global objectives, and (2) the current technological monologue can move toward a dialogue of diversified, just, and equitable technology. However, "sharing" implies mutual understanding and respect. Resilient, agile, socio-technological systems, from a cybernetics perspective, are those that develop and integrate relational intelligence and systemic consciousness; this conceptualization is central to the Indigenous sense of place.

### Future Work

There is no panacea to achieve environmental and social justice. As efforts toward these ends continually converge, GeoSemantics and socio-technological processes like Cybercartography may help to support an evolving technological framework that interconnects society, culture, environment, and technology. We hope that by giving due credit to

the Indigenous-led decolonization of AI and other digital technologies and amplifying the ideas of other such historically marginalized authors, we act as allies for a more just and inclusive technological and discursive space. We acknowledge that this advocacy may be both helpful and harmful, and thus, we strive to be self-reflective to ameliorate reinforcing Western-based institutional knowledge at the expense of other ways of knowing. With an open participation in science and policy, where Indigenous land-based communities map out relational meanings embedded ecological relationships, we suggest that this might forge *new, collaborative maps* for the direction of humanity, and we all become map-keepers. This paper suggests that gesturing toward a truly transdisciplinary and collaborative evolution in geospatial technology-related fields of Western science can help meet the challenges of addressing the SDGs while also addressing greater global justice.

This review and synthesis serves as a call to move toward developing more than a data infrastructure, but a critical and relevant knowledge infrastructure, where communities guide national and international policy, and where end users are the beginning of the process, as a principle. Given the ethical principles and policy documents described in Section 3.3 and the community-based processes described in Section 3.1, the development of a relevant and just knowledge infrastructure might begin locally. The use cases described in this review yield foundational frameworks for the processes with which researchers can connect with local Indigenous communities. These studies also highlight the researcher's responsibility to become aware of data sovereignty principles and traditional protocols that may be present in the territories on which we work, and to investigate local knowledge and approaches.

Systems thinking acknowledges that collective planetary thinking requires diverse worldviews. Technology development in a systems approach would include the worldviews of Indigenous peoples. Future work in these fields require an ever-evolving approach that includes participatory science and interdisciplinary and applied research to provide knowledge and capacity exchange opportunities with Indigenous peoples. The democratization of geospatial platforms and tools has proven effective in mediating between peoples of differing knowledge systems, building non-specialist technological capacity, and enabling community participation and action [16,74]. In this way, knowledge co-production using geospatial technologies are approaches to decentralize Western science as a primary way of knowing. A growing global Geo-verse, where no one is left behind, requires equitable access, participation in scientific and governance systems, the acknowledgement and integration of worldviews, and a depth of critical understandings of IK, as well as adherence to principles and protocols as outlined by communities.

Finally, we note that a cybernetics framework suggests that it is worthwhile to move forward with creating the conditions for new data infrastructures, critical and relevant knowledge infrastructures, and new futures. Complex systems produce emergent consequences that are impossible to predict. Thus, it is impossible to predict the impact of even one 'node', group, or human in the system. This argument insists that we collectively continue to do everything to work together toward positive outcomes for all.

**Author Contributions:** Conceptualization and writing, Christy M. Caudill; conceptualization and review and editing, Peter L. Pulsifer, D. R. Fraser Taylor and Romola V. Thumbadoo. All authors have read and agreed to the published version of the manuscript.

**Funding:** This research received no external funding.

**Acknowledgments:** We thank the many individuals and communities who have helped shape our knowledge. The IK holders that the authors have interacted with have expanded our understandings of these topics, and much more. Their wisdom, generosity, and patience form the bedrock of our collaborative research frameworks and programs.

**Conflicts of Interest:** The authors declare no conflicts of interest.

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
