# Peer review of "Meeting the Challenges of the UN Sustainable Development Goals through Holistic Systems Thinking and Applied Geospatial Ethics"

_ijgi, doi:10.3390/ijgi13040110_

Round 1

Reviewer 1 Report

Comments and Suggestions for Authors

The paper is very relevant and timely as the authors address several important topics including AI, Indigenous Knowlege, Systems Thinking, geospatial ethics, and the Sustainable Development Goals. Although I have read papers that address each topic separately, the authors are commended for demonstrating the intersection between all of these topics. I find the paper to be intellectually stimulating, thoroughly cite d, and well-written. My only suggestion for improvement is to consider inclusion of additional figures to add more visual support to the manuscript (currently just one figure in total). For example, could a figure or two be added for the Cybercartographic Atlases described in Section 3?

Author Response

The authors are grateful for the comments. We have added a figure in Section 3 for an example of a Cybercartographic Atlas.

Reviewer 2 Report

Comments and Suggestions for Authors

1.  This is a very well researched, substantive, and thorough tour-de-force major research paper by the authors and I salute them all for their work.  They take great care in acknowledging the sensitivity of the topic and their own background.  They provide ample opportunity for other researchers to take the many strands they lay out for further research.

2.  Page 1 line 38 - should tool STOCK be replaced by tool STACK?

3.  Page 3 I appreciate the paragraph on the term western - and the sensitive way the authors frame this.

4.  Page 4 - here is the goal - the goal gets a bit buried in the extensive and wonderful text, but perhaps it could be re-stated elsewhere in the paper:   As non-Indigenous scholars, we hope that this paper serves to foster informed, productive, and respectful dialogues so that the strength of diverse knowledges might offer whole systems approaches to decision-making that tackle wicked problems.

5.  Page 5 - You’re not stuck in traffic—you are traffic.”  <<--a very useful quote - could the authors expand a bit?

6.  Page 8 - semantics - that computers do not operate with an understanding <<-- extremely relevant - could be expanded... that WE assign meaning to the data; the computer or the tools do not do this.  Hence... our responsibility...for ethical decision making and beyond.

7.  Page 8 - Statement that - Geocybernetics development has focused on the scale of community data and knowledge.  <-- Perhaps a tiny overstatement?  The authors may be focusing on the geocybernetics that IS focused on community scale, but is there research outside this scale?  Is that research also informative?  And -- geocybernetics is a relatively new term, so ... caution is perhaps needed here.

8.  Page 16 - As DePuy et al. (2022) question: What does outsider advocacy for a plural reconceptualization look like, while avoiding romanticization or the continued privileging of outsider perspectives? And, "is it possible to contribute without falling into the trap of reinforcing western-based institutional knowledge at the expense of marginalized and subjugated ontologies?” <-- very appropriately mentioned and included, and appreciated by the reader.

9. Ethics section - there have been several significant ethics codes and articles written in the past 2 years around GIS that the authors might want to mention.

Author Response

We very much appreciate the reviewer's time. The comments substantially improved the manuscript.

  1. This is a very well researched, substantive, and thorough tour-de-force major research paper by the authors and I salute them all for their work.  They take great care in acknowledging the sensitivity of the topic and their own background.  They provide ample opportunity for other researchers to take the many strands they lay out for further research.
  2. Page 1 line 38 - should tool STOCK be replaced by tool STACK?

The text has been edited.

  1. Page 3 I appreciate the paragraph on the term western - and the sensitive way the authors frame this.
  2. Page 4 - here is the goal - the goal gets a bit buried in the extensive and wonderful text, but perhaps it could be re-stated elsewhere in the paper:   As non-Indigenous scholars, we hope that this paper serves to foster informed, productive, and respectful dialogues so that the strength of diverse knowledges might offer whole systems approaches to decision-making that tackle wicked problems.

We appreciated this useful comment, and the text has been edited to reflect this.

  1. Page 5 - You’re not stuck in traffic—you are traffic.”  <<--a very useful quote - could the authors expand a bit?

The text has been edited to more clearly and succinctly articulate this idea.

  1. Page 8 - semantics - that computers do not operate with an understanding <<-- extremely relevant - could be expanded... that WE assign meaning to the data; the computer or the tools do not do this.  Hence... our responsibility...for ethical decision making and beyond.

The text has been edited.

  1. Page 8 - Statement that - Geocybernetics development has focused on the scale of community data and knowledge.  <-- Perhaps a tiny overstatement?  The authors may be focusing on the geocybernetics that IS focused on community scale, but is there research outside this scale?  Is that research also informative?  And -- geocybernetics is a relatively new term, so ... caution is perhaps needed here.

We thank the reviewer, and made edits.

  1. Page 16 - As DePuy et al. (2022) question: What does outsider advocacy for a plural reconceptualization look like, while avoiding romanticization or the continued privileging of outsider perspectives? And, "is it possible to contribute without falling into the trap of reinforcing western-based institutional knowledge at the expense of marginalized and subjugated ontologies?” <-- very appropriately mentioned and included, and appreciated by the reader.
  2. Ethics section - there have been several significant ethics codes and articles written in the past 2 years around GIS that the authors might want to mention.

The comment is noted, and we have added to the referenced codes and articles.

Reviewer 3 Report

Comments and Suggestions for Authors

Thank you for submitting your work. My overall critique is as follows:

Title & Abstract

i.            The abstract is excessively lengthy, making it difficult for comprehensive reading. Rather than incorporating citations from other works, it should serve as a platform to showcase the accomplishments derived from the present study.

ii.            The objectives of the study lack clarity, with only a few sentences mentioning the author's intentions, such as "We will discuss semantic mediation frameworks..." and "This paper reviews approaches like Cybercartography as an avenue to address gaps..." These statements hint at the author's intentions but fail to articulate specific objectives.

iii.            Furthermore, the abstract lacks a description of the applied methodology; the mere mention of "this paper reviews" lacks specificity and purpose. It is essential to clarify the nature of the review being conducted.

iv.            Moreover, a representation of the review's results is missing in the abstract, leaving readers in the dark about the findings of the study.

v.            It is imperative to provide a clear explanation of the types of Sustainable Development Goal (SDG) challenges that, this review article aims to meet. Introduction

i.            To enhance clarity in understanding, rephrasing many sentences is required. For instance, rephrase the following sentence for improved comprehension: “Although positioning the voices of Indigenous peoples with a holistic place-based understanding is a key intent of the research, we couch our own understanding of holistic Systems Thinking in a western framework of Cybernetics {e.g., 20, 21, 22, 23]. building on the seminal formalization of Geocybernetics by Reyes et al [24, 25]..”

ii.            Similarly, rephrase the sentence “Figure modified after Reyes et al. (2014), Figure 3.1 [24] ”to seamlessly integrate it into the text.

iii.            The current introduction lacks a smooth transition between concepts. There is jargon of ideas included in the introduction that is impeding the reader's ability to grasp the author's intentions.

iv.            Elaborate in detail on the specific challenges related to Sustainable Development Goals (SDGs) that this article aims to address. Provide a comprehensive discussion of geospatial ethics to familiarize readers with these terms.

v.            Moreover, the introduction should explicitly highlight the gaps in existing literature, articulate why current approaches fall short in addressing these gaps, and explain how the proposed review study, employing Holistic Systems Thinking and Applied Geospatial Ethics, intends to bridge these holes. This detailed exposition will provide readers with a clear understanding of the study's objectives and significance.

 Discussion

i.            Sentence restructuring is necessary in this section. For instance, consider rephrasing the following sentence for improved clarity: "It is said that everything happens somewhere, assessing environmental, social, and statistical data through geospatial data, tools, and technologies provides a powerful and proven, approach to offer immediate solutions for adaptation or timely crisis mitigation”.

ii.            Additionally, this section is spoiled by the inclusion of multifaceted ideas and lacks a seamless transition from one concept to another, as well as from one paragraph to the next.

iii.            A synthesis of the literature is evidently absent; the authors could benefit from organizing the main ideas and extracts into a table or/and figure to enhance understanding.

iv.            Furthermore, the section fails to discuss any Sustainable Development Goal (SDG). If the review encompasses all SDG goals, the authors should elaborate on this comprehensively, providing a detailed explanation. This would greatly improve the overall coherence of the section and provide readers with a clearer understanding of the review's scope and objectives.

Conclusions

i.            The authors have not presented any synthesis of the achievements of the review study. A vital part of any conclusion section of the article is indeed inclusion of highlights of the significant contributions made by the study.

ii.            Additionally, incorporating policy implications of the findings will provide a practical dimension to the research., which is currently absent in the conclusion part.

iii.            Furthermore, it is beneficial to include one or two paragraphs related to future work, outlining potential avenues for further research.

iv.            Add the mention of any types of limitations you face during the conduction of this review.

Comments on the Quality of English Language

Moderate editing required

Author Response

The authors are most appreciative for this thorough review. There was much that needed clarification. Although many things needed simplifying, we recognize that the holistic nature of Systems Theory is multifaceted and transdisciplinary, and have added text to help with the flow from one concept to the next. Again, many thanks to the reviewer. The manuscript is much improved.

Title & Abstract

  1. The abstract is excessively lengthy, making it difficult for comprehensive reading. Rather than incorporating citations from other works, it should serve as a platform to showcase the accomplishments derived from the present study.

Thank you, and this valuable comment was well-received. We cut down the abstract, but importantly, reshaped and reworded it such that it spoke to many of the comments below for the Abstract section.

  1. The objectives of the study lack clarity, with only a few sentences mentioning the author's intentions, such as "We will discuss semantic mediation frameworks..." and "This paper reviews approaches like Cybercartography as an avenue to address gaps..." These statements hint at the author's intentions but fail to articulate specific objectives.

We have edited the Abstract to more explicitly state the aims of the paper and what this study sought to do.

iii.            Furthermore, the abstract lacks a description of the applied methodology; the mere mention of "this paper reviews" lacks specificity and purpose. It is essential to clarify the nature of the review being conducted.

We have edited the text and added pointed statements with regard to the nature of the review,, such as “This paper reviews transdisciplinary, holistic, and participatory approaches…” then, “Thus, this paper will discuss how holistic systems thinking” and ended with, “This review will investigate the inequality in technological systems…”

  1. Moreover, a representation of the review's results is missing in the abstract, leaving readers in the dark about the findings of the study.

We have edited the text to better define the outcomes of the review and synthesis, both in the Abstract and in the Conclusions.

  1. It is imperative to provide a clear explanation of the types of Sustainable Development Goal (SDG) challenges that, this review article aims to meet. 

We have noted that the challenges are the approaches themselves, and the current use of geospatial technologies to assess, track, and support meeting those goals. 

The authors note: “Geospatial technologies have proven indispensable in assessing and tracking fundamental components of each of the 17 SDGs, including climatological and ecological trends and changes and humanitarian crises and socio-economic impacts.”

Introduction

  1. To enhance clarity in understanding, rephrasing many sentences is required. For instance, rephrase the following sentence for improved comprehension: “Although positioning the voices of Indigenous peoples with a holistic place-based understanding is a key intent of the research, we couch our own understanding of holistic Systems Thinking in a western framework of Cybernetics {e.g., 20, 21, 22, 23]. building on the seminal formalization of Geocybernetics by Reyes et al [24, 25]..”

Noted, with thanks. This and many sentences were restructured.

  1. Similarly, rephrase the sentence “Figure modified after Reyes et al. (2014), Figure 3.1 [24] ”to seamlessly integrate it into the text.

iii.            The current introduction lacks a smooth transition between concepts. There is jargon of ideas included in the introduction that is impeding the reader's ability to grasp the author's intentions.

We have made note of this (helpful) comment, and moved sentences and paragraphs. We have set up the end of the Introduction as its own section to set up the necessary concepts in Systems Thinking to move on to a discussion of them in the following sections. This is concept-heavy, but we have edited to take greater care with the readability. We have also added in transitions.

  1. Elaborate in detail on the specific challenges related to Sustainable Development Goals (SDGs) that this article aims to address. Provide a comprehensive discussion of geospatial ethics to familiarize readers with these terms.

We have reworded to state that the challenge is the inherent, relational and interconnected nature of the SDGs, which are not served by further reducing the complexity. We argue that, to the contrary, a “deeper, more comprehensive understanding of the complex and multi-factorial challenges delineated in the SDGs. The immensity of the challenged outlined in the SDGs calls for inclusive, and holistic approaches, coupled with transformative uses of geospatial and other digital technologies.” We have edited the text, throughout, to be more clear about the broad conceptual approach to meeting the challenges outlined. As there are many concepts in this paper already, and it is quite long, we do not think it serves the paper to enumerate the SDGs.

  1. Moreover, the introduction should explicitly highlight the gaps in existing literature, articulate why current approaches fall short in addressing these gaps, and explain how the proposed review study, employing Holistic Systems Thinking and Applied Geospatial Ethics, intends to bridge these holes. This detailed exposition will provide readers with a clear understanding of the study's objectives and significance.

We have added, “Many have argued that adjustments to existing policies and attempting market-driven changes are inadequate or ill-positioned to deliver the necessary changes in how people, globally dominant cultures and consequently, global systems, sustainably relate to and draw benefits from the environment [3, 4, 5, 6]. In this paper, we will discuss how holistic systems thinking—societal, political, environmental, and economic choices are never separate, but considered as a system—might offer approaches to decision-making that tackle wicked problems”. We have edited the text, throughout, to help with readability, and to more clearly articulate the objectives and significance.

 Discussion

  1. Sentence restructuring is necessary in this section. For instance, consider rephrasing the following sentence for improved clarity: "It is said that everything happens somewhere, assessing environmental, social, and statistical data through geospatial data, tools, and technologies provides a powerful and proven, approach to offer immediate solutions for adaptation or timely crisis mitigation”.

Noted, and the text has been edited.

  1. Additionally, this section is spoiled by the inclusion of multifaceted ideas and lacks a seamless transition from one concept to another, as well as from one paragraph to the next.

Noted, with thanks. Editing the text and adding in transitions has helped make the arguments more congruent, and the text more readable.

iii.            A synthesis of the literature is evidently absent; the authors could benefit from organizing the main ideas and extracts into a table or/and figure to enhance understanding.

As for a table of literature and resources, we decided that tables did not lessen the complexity, but added to it (with an already very long paper). We have instead edited the text, added references, and integrated references from various fields where appropriate. This has helped the text considerably. We have used this comment to edit for overall coherence and readability of the paper.

  1. Furthermore, the section fails to discuss any Sustainable Development Goal (SDG). If the review encompasses all SDG goals, the authors should elaborate on this comprehensively, providing a detailed explanation. This would greatly improve the overall coherence of the section and provide readers with a clearer understanding of the review's scope and objectives.

These comments are well-received, and, as noted above, we have edited the text throughout to help with readability, and to more clearly articulate the objectives and significance. Again, these comments have helped make the paper much better.

Conclusions

  1. The authors have not presented any synthesis of the achievements of the review study. A vital part of any conclusion section of the article is indeed inclusion of highlights of the significant contributions made by the study.

These comments helped reshape the Conclusions section, and were well-received.

  1. Additionally, incorporating policy implications of the findings will provide a practical dimension to the research., which is currently absent in the conclusion part.

We agree that policy recommendations are important, but decided that within the scope of this paper, it was best to keep the focus on practical implementation of research with communities. We have expanded on this throughout the paper.

iii.            Furthermore, it is beneficial to include one or two paragraphs related to future work, outlining potential avenues for further research.

The Conclusion was much better served by breaking it up, and a Future Work section helped with this. We considered limitations in this section as well.

Round 2

Reviewer 3 Report

Comments and Suggestions for Authors

Thank you for addressing my concerns to enhance quality of the manuscript.

Comments on the Quality of English Language

Minor editing required